# A Design Approach to Reducing Stress and Distortion Caused by Adhesive Assembly in Micromachined Deformable Mirrors

**DOI:** 10.3390/mi14040740

**Published:** 2023-03-27

**Authors:** Wenkuan Man, Thomas G. Bifano

**Affiliations:** 1Division of Materials Science and Engineering, Boston University, Boston, MA 02215, USA; 2Mechanical Engineering Department, Boston University, Boston, MA 02215, USA; 3Photonics Center, Boston University, Boston, MA 02215, USA

**Keywords:** deformable mirror, micromachines, electromagnetic actuation, packaging and assembly, thin-film stress

## Abstract

A common problem in deformable mirror assembly is that the adhesion of actuators to an optical mirror face sheet introduces unwanted topography due to large local stresses generated at the adhesive joint. A new approach to minimizing that effect is described, with inspiration taken from St. Venant’s principle, a fundamental precept in solid mechanics. It is demonstrated that moving the adhesive joint to the end of a slender post extending from the face sheet largely eliminates deformation due to adhesive stresses. A practical implementation of this design innovation is described, using silicon-on-insulator wafers and deep reactive ion etching. Simulation and experiments validate the effectiveness of the approach, reducing stress-induced topography on a test structure by a factor of 50. A prototype electromagnetic DM using this design approach is described, and its actuation is demonstrated. This new design can benefit a wide range of DMs that rely on actuator arrays that are adhesively bonded to a mirror face sheet.

## 1. Introduction

Deformable mirrors (DMs) are used in adaptive optics (AO) systems for imaging, communication, and beam forming [1]. The essential function of a DM is to provide a reflective surface of high optical quality that can be shaped by an underlying array of actuators to compensate for undesired aberrations in an optical system.

Architectures for DM design are driven by the optical system’s technical performance requirements, including actuator stroke, actuator count, actuator response time, actuator repeatability, mirror surface quality, mirror reflectivity, and mirror diameter. The first DMs, developed in the 1970s, were used for defense applications and for ground-based large-aperture telescope imaging. Most of those DMs were deformed using arrays of piezoelectric actuators bonded to the back of centimeter-thick monolithic mirror face plates [2]. Considerations of cost have limited the more widespread use of DMs in AO applications until the introduction of micromachined devices [3]. Microelectromechanical system (MEMS) foundry processes provided an economical and scalable way of producing DMs suitable for applications of adaptive optics in astronomy, vision science, laser communication, and microscopy. An enabling feature of surface micromachined DMs is that the small gaps achievable with that fabrication approach allowed the practical use of electrostatic and electromagnetic actuation [4,5,6,7,8,9,10].

The most common DMs in use today comprise continuous reflective face sheets supported by underlying arrays of surface-normal actuators. The face sheet can be fabricated integrally with the actuator array using multilayer thin-film silicon surface micromachining, or it can be bonded to the actuator array directly in an adhesive assembly process. An advantage of surface micromachining is that the actuator and mirror layers can be produced entirely in silicon through refractory deposition processes, reducing or eliminating the need for precision assembly and the stresses associated with adhesive bonding. A disadvantage is that surface micromachining generally requires etch access holes in the mirror face sheet layer to allow sacrificial etching of the underlying layers associated with the actuator array. In many applications, this disadvantage is negligible. However, in some applications such as high-energy laser (HEL) beam propagation and solar astronomy, small defects such as etch access holes in the DM face sheet are intolerable. In those applications and others, adhesive bonding between the continuous face sheet and the actuator array is the main technique used for DM assembly.

Because the DM face sheet is compliant, complex stress fields induced by its adhesion to the actuator array can introduce unintended deformation. Some of that deformation occurs at spatial frequencies within the control band of the DM. For example, if adhesion to the actuator array causes static curvature to change across the entire DM, the actuators themselves may be able to compensate for that curvature through active control, albeit at the cost of some of the available DM stroke that could otherwise be used for aberration compensation. However, in practice, most of the deformation due to actuator adhesion occurs locally in the regions of the adhesion, producing high-spatial-frequency shape errors on the face sheet that cannot be compensated for by the actuators. That undesired topography sets a floor for the DM’s achievable wavefront error reduction in an adaptive optics system.

The widely used Marechal criterion can be used to assess whether the magnitude of uncorrected face sheet deformation will adversely affect optical performance of the DM. That criterion is that an optical system can be considered “well-corrected” if its residual root-mean-square wavefront error is less than λ/14, where λ is the system’s optical wavelength of interest. Because of reflection, residual topography errors on the DM are doubled in their impact on residual system wavefront error. Consequently, we can consider uncorrectable high-spatial-frequency topography on a DM surface to be acceptable if it has a root-mean-square deviation from flatness of less than λ/28. For example, in a visible imaging application at a wavelength of ~650 nm, this would require the DM face sheet uncorrectable topography to be less than 23 nm-rms.

Many DMs have been produced by bonding piezoelectric actuators [11,12,13] or permanent magnets [14,15] to the face sheet. Various techniques and materials have been used to provide a reliable, stable, low-stress adhesive connection between components. Inevitably, residual stresses form in adhesive joints due to the stiffening, viscosity change, and shrinkage that generally accompanies adhesive curing. Peak residual stresses in cured adhesive joints can exceed tens of megapascals [16,17].

Stress states associated with adhesive joints are difficult to model, although there have been some experimental and numerical results that point to best practices for limiting stress and deformation associated with adhesive joints [16,18,19]. Reducing deformation due to stresses in microfabricated devices can be achieved by choosing adhesives with lower modulus of elasticity, an approach that is commonly used for MEMS die attachment and packaging. However, for attaching actuators to face sheets in DMs, this approach has limitations; flexibility in the adhesive joint reduces achievable stroke and can lead to adverse long-term effects such as creep. Alternatively, the magnitude of stress-induced deformation due to adhesive bonding can be reduced by using a thicker face sheet. Again, this approach limits achievable stroke for the actuators due to the increased stiffness of a thicker face sheet.

In this paper, we propose a new design approach for reducing the detrimental effects of adhesion-induced stresses on face sheet topography. The approach does not compromise stiffness of the adhesion joint and does not require thickening of the face sheet. We demonstrate the effectiveness of this new design approach through design, modeling, and prototype production of an electromagnetically actuated deformable mirror.

## 2. Materials and Methods

All silicon-based samples were fabricated using silicon-on-insulator (SOI) wafers (Ultrasil LLC, Hayward, CA, USA). Photoresist lithography was used to pattern silicon-on-insulator (SOI) wafers with a device layer thickness of 10 µm (AZ 10XT, MicroChemicals GmbH, Ulm, Germany). Deep reactive ion etching (DRIE) processes were conducted on an SPTS Rapier (Omega LPX Rapier, Orbotech Ltd., Yavne, Israel). Micro magnets (Cyl0003-50, SuperMagnetMan Company, Pelham, AL, USA) and ultraviolet-curable adhesive (NOA 81, Norland Inc., Jamesburg, NJ, USA) were used for attachment processes.

Fabricated sample surface topography was measured using a surface mapping interferometer (NewView 9000, Zygo Corporation, Middlefield, CT, USA). Power supplies (72-2685 Digital control DC power supply, TENMA Corporation, Tokyo, Japan) were used as the power source for electromagnetic actuation tests.

## 3. Design and Simulation

The core of the new design approach is based on St. Venant’s principle, i.e., high-order moments of mechanical loading decay rapidly for regions far from those complex stresses and stress concentrations. The adhesion boundary itself will inevitably result in complex stresses and stress concentrations, but if that boundary can be moved far from the face sheet (i.e., a distance larger than the width of the adhesive joint), then the resulting stress on the face sheet will be substantially reduced in magnitude. In static equilibrium, the resultant forces and torques at the face sheet reflect those in the adhesion layer. If the adhesion layer cross-sectional geometry is simple and symmetric (e.g., circular or square) and is moved far away from the face sheet, then the resultant stresses on the face sheet will be small and mainly surface-normal, producing small deformations that can easily be compensated for by the actuators themselves. Figure 1 illustrates the concept for a simplified example: a post with a remote adhesive joint suspended from a rigid support. Stresses at the support plane are greatly reduced from those at the adhesive joint.

What makes this design approach feasible is the commercial availability of a MEMS process, deep reactive ion etching (DRIE), through which a nearly perfectly flat, polished, single-crystal silicon face sheet can be formed simultaneously with an integral array of slender, single-crystal posts. Such a structure can be made from one monolithic single-crystal silicon wafer or from a silicon-on-insulator (SOI) wafer comprising two single-crystal wafers (a thin device layer and a thick handle layer) joined together by a thin layer of thermally grown silicon dioxide. If one patterns and etches the handle layer using DRIE, it is relatively easy to produce a face sheet, a frame, and an array of integral posts. Bond attachments of actuators to the distal ends of the posts, even if highly stressed, will not relay higher moments of force to the face sheet. Consequently, one can expect the face sheet of the assembled device to exhibit lower stress-induced surface deformation than if the actuators were bonded directly to a comparably thick face sheet alone.

To validate the effectiveness of the new design approach in reducing stress-induced face sheet deformation, rotationally symmetric numerical simulations of an idealized single-actuator DM were conducted using COMSOL finite element software (version 6.1). The face sheet was modeled as a cylindrical disc with a diameter of 3 mm and thickness of 10 µm, using material properties of single-crystal silicon. Its edges were fixed. The adhesion layer was modeled as a cylindrical disc with a diameter of 300 µm and thickness of 5 µm, using material properties of a UV-curable polyurethane-based optical adhesive (Norland Optical Adhesive 81). A radially symmetric tensile stress of 5 MPa was imposed on the upper surface of the adhesive disc.

Two scenarios were simulated. In the first, the adhesive layer was connected directly to the face sheet. In the second, a long slender post was inserted between the adhesive layer and the face sheet. The post was modeled as a cylindrical disc with a diameter of 300 µm and height of 500 µm, using material properties of single-crystal silicon. Schematic cross-sections (not to scale) of the two simulated cases are illustrated in Figure 2.

The resulting mirror face sheet surface distortion at equilibrium for the two simulated scenarios are shown in Figure 3. For the simulation in which the adhesion was in direct contact with the face sheet, the face sheet experienced peak Von Mises stresses of ~170 MPa, which caused a prominent, highly localized deformation having a peak magnitude of 2.2 µm and deviation from flatness over the face sheet of 626 nm-rms. For the simulation in which the adhesion was moved to the bottom of a slender post connected immediately below the face sheet, the face sheet experienced peak Von Mises stresses of <1 MPa, which resulted in peak simulated face sheet deformation of 0.01 nm and deviation from flatness over the face sheet of 0.002 nm-rms, many orders of magnitude lower than the direct-attachment case.

The result of these simulations suggests that a significant improvement in DM face sheet flatness can be achieved by displacing the actuator array adhesion joints to the distal ends of slender posts. In Section 4, we experimentally validate this expectation on a test structure.

## 4. Experimental Results

To experimentally validate the advantage of the proposed design approach, a test structure was made. In this test structure, two small permanent magnets were adhesively bonded to a single face sheet: one at the distal end of a slender integral post and one directly onto the face sheet.

The test structure comprised an 18 mm square silicon device fabricated from an SOI wafer. The SOI device, polished on both sides, featured a 10 µm thick device layer, a 1 µm thick buried oxide layer, and a 600 µm thick handle layer. The handle layer was patterned lithographically and etched all the way through to the buried oxide layer using DRIE, leaving a 10 µm thick, 12 mm diameter circular face sheet supported by 600 µm thick fixed edges at its perimeter and one 300 µm diameter, 600 µm tall circular post attached off-center to the face sheet. The lithographic pattern also included holes in the square frame to facilitate subsequent alignment of electromagnetic actuator coils. After DRIE etching, the exposed areas of silicon dioxide were removed through a wet HF etching process. The silicon dioxide between the post and the face sheet remained intact. An engineering sketch of the test structure is shown in Figure 4. An SEM image of the post attachment to the face sheet is shown in Appendix A.

Surface topography maps of the top of the face sheet were made using a surface mapping interferometer (Zygo NewView 9000) in two regions: one above an area of the face sheet to which a micromagnet magnet would be attached subsequently and one above the integral off-center post to which a magnet would be attached subsequently. These maps are shown in Figure 5. The face sheet on the fabricated test structure was found to be relatively flat and smooth. In the 1.54 mm square region corresponding to the location where a magnet would be subsequently adhesively bonded, the root-mean-square deviation from flatness was measured to be 8 nm-rms. In a similar-sized square region directly above the post some initial deformation of the face sheet was observed. For this region the root-mean-square deviation from flatness was measured to be 21 nm-rms. Since the post and the face sheet were formed from identical single-crystal silicon materials, this deformation was probably due to residual stresses in the 1 µm thick silicon dioxide layer between the post and the face sheet. Those stresses were not considered in the previous simulation.

Using a custom-built precision alignment apparatus and assembly process, two permanent neodymium magnets (300 µm in diameter and 500 µm tall) were subsequently attached to the test structure using a low-viscosity UV-curable adhesive (Norland Optical Adhesive 81). The adhesive was cured in UV light for 90 min, and then thermally cured at 90 °C on a hotplate for 10 h. One magnet was attached to the distal end of the post, and the other was attached to the face sheet directly below the region depicted in Figure 5a. A photo of the test structure with magnets attached to the post and to the bottom of the face sheet is shown in Figure 6.

After curing the adhesive, surface topography maps of the top of the face sheet of the test structure were made in the same two regions shown in Figure 5. In the area of the face sheet above the directly attached magnet, a substantial increase in topography was observed. Peak magnitude of deformation was more than 2 µm, and root-mean-square deviation from flatness was 317 nm-rms. In the area of the face sheet above the magnet attached to the post, the root-mean-square deviation from flatness was 29 nm-rms, 8 nm-rms larger than it was before the magnet was adhered to the post. The results are shown in Figure 7.

To extend these test structure experiments, we produced a 12 mm diameter, 10 µm thick, edge-supported face sheet with an array of 37 posts, spaced 1.5 mm apart on a hexagonal grid. We measured surface topography of the face sheet and compared it to a face sheet that was fabricated without the post array. The results are shown in Figure 8. The overall deviation from flatness of the two edge-supported face sheets is comparable and well within the range of compensation for a typical DM actuator. The local topographic deviation from flatness over a 1.5 mm square area was 7 nm-rms for the face sheet without posts and 28 nm-rms for the face sheet with posts, comparable to the data previously shown in Figure 5.

Magnets were attached to the 37 posts using UV-curable adhesive. The face sheet deformed considerably after magnet attachment and adhesive curing. A photograph of the magnets attached to the face sheet with posts is shown in Figure 9, along with a measured topography map of the face sheet. Significant deformation of the face sheet was observed, measuring almost 5 µm in a nearly spherical depression with hexagonal boundaries. The cause of this depression was the magnetic repulsive force that each of the magnets exerted on its neighbors. All magnets were oriented in the same direction. Consequently, they repelled one another. For magnets near the center of the array, this force was balanced by symmetrically located near neighbors. However, for the magnets forming the outer ring of the hexagonal array, the repulsive force was unbalanced, leading to a torque of these peripheral magnets that was relayed to the posts and bent the face sheet at the array periphery. A solution to this problematic magnet-induced deformation would be to mount an extra ring (or two) of magnets in the array, with the outermost ring of magnets bonded to the rigid device frame.

Despite the substantial curvature induced by magnet attachment, we tested DM actuation. We assembled the face sheet with attached posts and magnets to an underlying structure comprising an array of coils that could, if energized by an applied current, exert an electromagnetic force on the magnets attached to the posts. A schematic of the assembly setup for electromagnetic actuation tests is shown in Appendix A. Each coil was fabricated from 75 µm diameter copper wire with 2 µm insulating sheathing. Coils comprised 16 windings of the wire in a compact structure: four planar spirals in each of four stacked layers. Microscope photographs of the coil are shown in Appendix A. Three coils were mounted in a common frame and positioned so that their top surface was located 30 µm below a corresponding magnet on the structure comprising the face sheet, posts, and magnets. The three coils that were actuated corresponded to three central magnets in the 37-post array, one beneath the center of the face sheet and two immediately adjacent to that central one on either side.

Coils were driven by three independent DC current sources capable of providing bipolar current up to ±500 mA. To differentiate topography due to actuation from topography due to the initial static deformation previously described, topographic maps were made by subtracting the unactuated surface map measurement from all actuated surface map measurements. Results are depicted in Figure 10.

## 5. Discussion

Avoiding undesired face sheet deformation in DM fabrication is important, particularly when that deformation occurs at high spatial frequencies that cannot be compensated for by the actuators themselves. In many applications, the processing step most prone to introducing such deformations is adhesive bonding of actuator structures to the face sheet. In this work, we demonstrated a technique to limit deformation-inducing stresses from adhesive joints by moving those stresses to a plane far from the face sheet surface. We employed a common micromachining process (deep reactive ion etching) and an inexpensive, super-polished raw material (silicon-on-insulator wafer) to produce face sheets that are relatively immune from adhesion-induced deformation. The test structures that we produced were of a scale (12 mm diameter) that is typical for DMs used in applications including ophthalmoscopy, microcopy, laser communications, and astronomical imaging. The approach we describe is scalable to smaller and larger DM sizes. It is applicable to DMs with either magnetic or piezoelectric actuators, and it increases the range of adhesives that are suitable for DM assembly, which previously required careful selection of low-stress adhesives that sometimes compromise bond strength and reliability.

## Figures and Tables

**Figure 1 micromachines-14-00740-f001:**
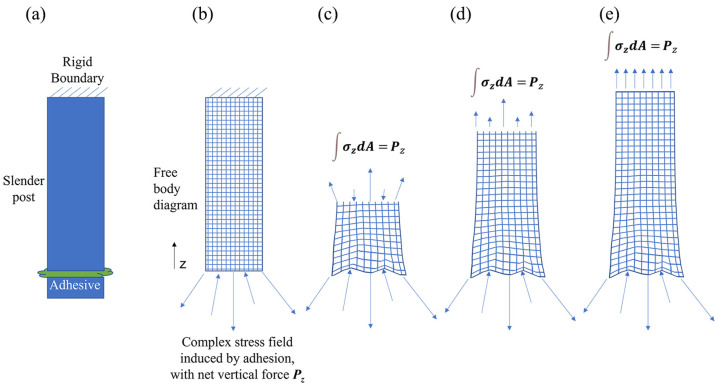
Schematic cross-section illustrating the proposed approach to mitigate complex stresses due to adhesion interfaces. (**a**) A post is suspended from a rigid boundary and has an adhesive interface some distance from the boundary. (**b**) A complex stress state with regions of stress concentration is shown in a free-body sketch of the bottom of the post. The net force is equal to the integral of stresses evaluated over the post cross-sectional area. (**c**) Near the adhesive interface, the resultant stresses on an interior cross-section of the post are still complex. (**d**) At interior cross-sections further away from the adhesive interface, the resultant stresses become more uniform. (**e**) Far from the plane of imposed adhesive stress, the stress state is nearly uniform. In all interior sections, the net force is equal and opposite the force at the interface, as required by static equilibrium.

**Figure 2 micromachines-14-00740-f002:**
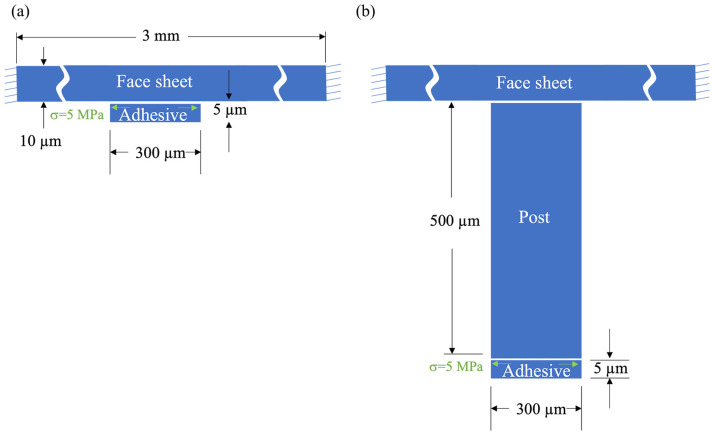
Schematic cross-sections (not to scale) illustrating the two cases simulated to demonstrate the effectiveness of the proposed approach. (**a**) A thin adhesive disc with 5 MPa radial outward stress imposed on its upper boundary was attached to the bottom of a thin, round circular face sheet. (**b**) The same adhesive disc was attached to the bottom of a slender silicon post extending from the bottom of the face sheet.

**Figure 3 micromachines-14-00740-f003:**
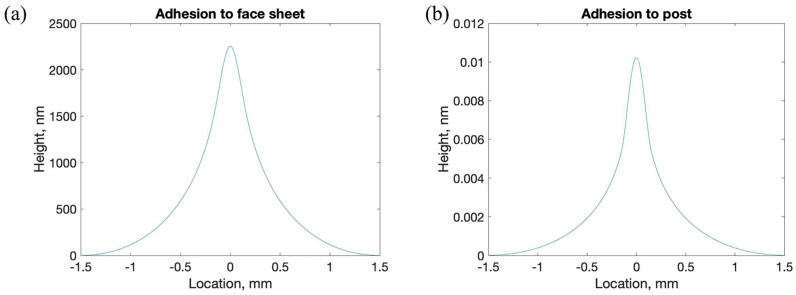
Simulation results for the new design approach, illustrating a substantial reduction in face sheet deformation due to adhesive stresses based on the location of the adhesive connection. (**a**) Simulated surface profile across the idealized single-actuator silicon DM face sheet (3 mm diameter, 10 µm thickness) with a stressed (5 MPa) adhesive joint (300 µm diameter, 5 µm thickness) directly connected to the bottom of the face sheet. Peak deformation was 2.2 µm, and deviation from flatness was 626 nm-rms. (**b**) Simulated topography of the same face sheet with the same adhesive joint connected to an integral silicon post (diameter 300 µm, height 500 µm) extending from the bottom of the face sheet. Peak simulated face sheet deformation was 0.01 nm, and deviation from flatness was 0.002 nm-rms.

**Figure 4 micromachines-14-00740-f004:**
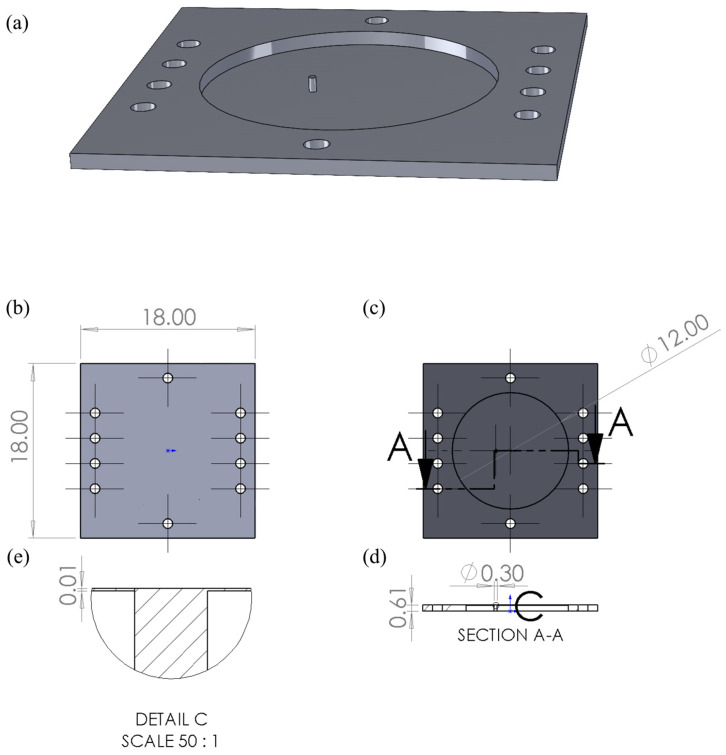
Perspective view and scale drawings of the test structure used for experimental validation. (**a**) An SOI waver was processed using lithography, DRIE etching, and wet etching to produce a square structure with a 12 mm diameter circular face sheet containing one off-center post. Alignment holes were etched into the square frame to help with subsequent magnet assembly. (**b**) Top-view scale drawing of the 18 mm square test structure. (**c**) Bottom-view scale drawing of the test structure. (**d**) Sectional view showing cross-section of the post. (**e**) Detailed enlargement of the post connection to the face sheet. Not shown is the 1 µm thick oxide layer between the post and the face sheet.

**Figure 5 micromachines-14-00740-f005:**
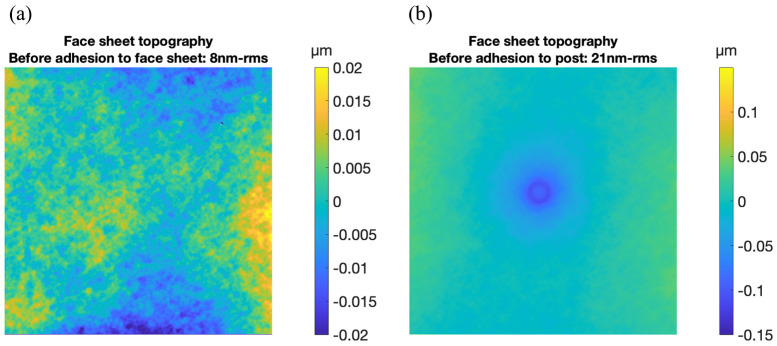
Surface topography maps of two regions of the test structure face sheet before adhesive attachment of magnets to face sheet or post. (**a**) In a 1.54 mm square region away from the area of post attachment, the root-mean-square deviation from flatness was 8 nm-rms. (**b**) In a 1.54 mm square region directly above the post attachment, some deformation of the face sheet due to the 300 µm diameter post attachment was evident. Root-mean-square deviation from flatness was 21 nm-rms.

**Figure 6 micromachines-14-00740-f006:**
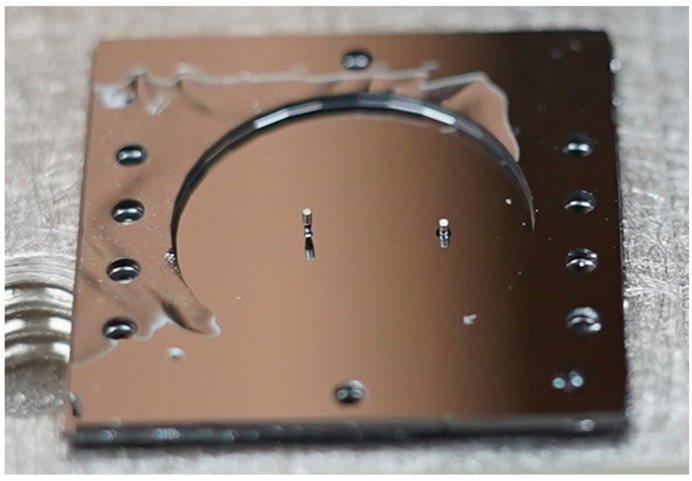
Photo of test structure with two magnets adhesively attached to the face sheet, one **(left**) at the end of a post, and one (**right**) directly on the face sheet.

**Figure 7 micromachines-14-00740-f007:**
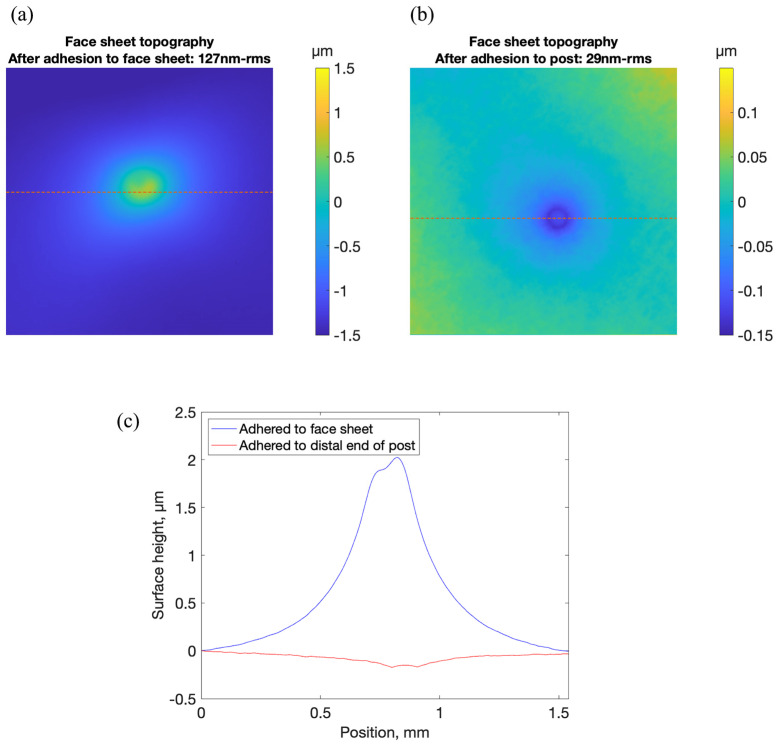
Surface topography maps of two regions of the test structure face sheet after adhesive attachment of two magnets: one to the face sheet and one to a post extending from the face sheet. (**a**) In a 1.54 mm square region above the face sheet in the vicinity of the direct magnet attachment, the root-mean-square deviation from flatness was 317 nm-rms. (**b**) In a 1.54 mm square region directly above the post attachment with magnet attached, the root-mean-square deviation from flatness was 29 nm-rms. (**c**) Face sheet deformation along indicated dotted lines in (**a**,**b**), demonstrating the effectiveness of stress reduction achieved through adhesion of the magnet to the distal end of the post.

**Figure 8 micromachines-14-00740-f008:**
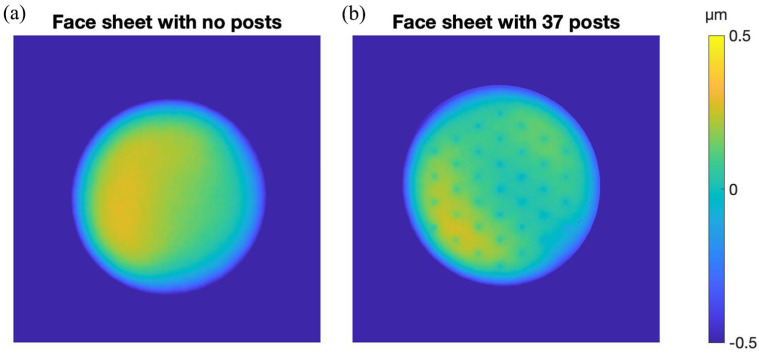
Surface topography maps of two 12 mm diameter edge-supported face sheets fabricated from the 10 µm thick device layer of a silicon-on-insulator wafer. (**a**) Face sheet with no posts. Deviation from flatness over the entire surface was 250 nm-rms. Local deviation from flatness in a 1.5 mm square region was 5 nm-rms. (**b**) Face sheet with 37 posts etched into the handle layer. Deviation from flatness over the entire surface was 180 nm-rms. Local deviation from flatness in a 1.5 mm square region was 29 nm-rms.

**Figure 9 micromachines-14-00740-f009:**
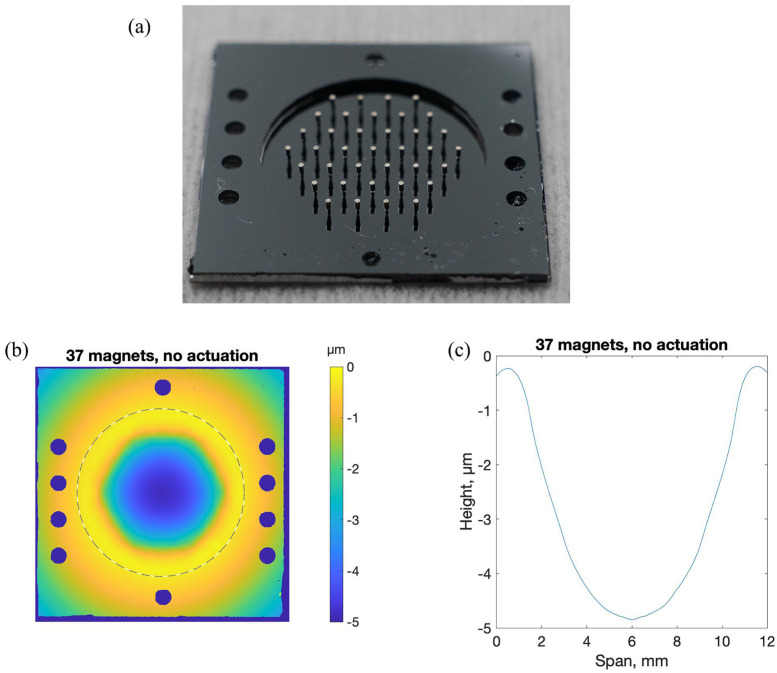
DM face sheet with posts and magnets attached. (**a**) Photo of SOI device showing the 18 mm square frame with guide holes for subsequent assembly with actuator electromagnetic coil array, the 12 mm diameter handle layer etch that exposed the 10 µm thick face sheet, and the array of 37 posts etched out of the handle layer and attached to the face sheet through a thin silicon dioxide layer. (**b**) Surface topography maps of the face sheet after adhesive attachment of magnets. The dashed line marks the boundary of the face sheet. Nearly 5 µm of deformation of the face sheet was observed, in a pattern bounded by the hexagonal shape of the periphery of the magnet array. (**c**) Plot of face sheet profile measured along the horizontal dotted line.

**Figure 10 micromachines-14-00740-f010:**
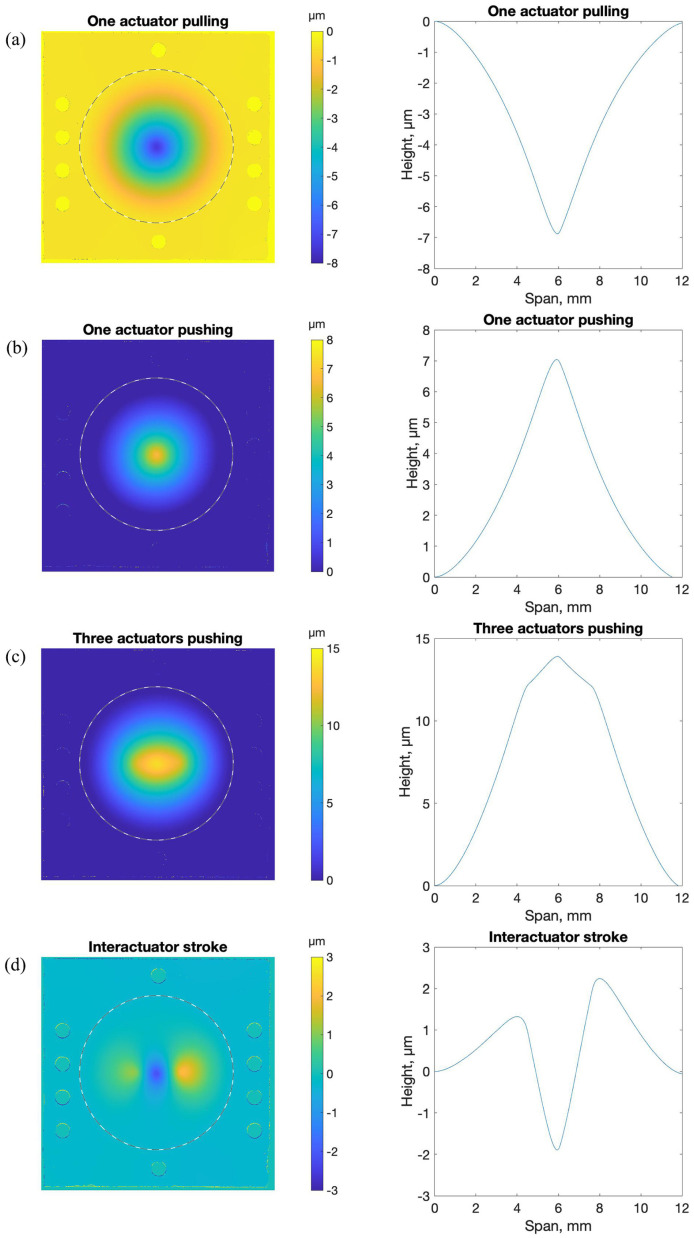
Measured topography due to electromagnetic actuation of a test device made with 10 µm thick, 12 mm diameter face sheet supported by an array of 37 posts, with a magnet adhesively bonded to the distal end of each post. Actuation was achieved by driving current through one or more of three copper coils spaced 30 µm from the base of three magnets near the center of the array. All topographic measurements are made by subtracting the measured topography from the static face sheet shape shown in the previous figure. Surface maps are shown on the left, with face sheet perimeter denoted by a dashed line. Horizontal profiles across the center of the map are shown on the right. (**a**) Center actuator energized to provide attractive force (pulling) at 500 mA current. Peak deflection ≈ 7 µm. (**b**) Center actuator energized to provide repulsive force (pushing) at 500 mA current. Peak deflection ≈ 7 µm. (**c**) Center three actuators energized to provide repulsive force (pushing) at 500 mA. Peak deflection ≈ 14 µm. (**d**) Center actuator energized to provide attractive force (pulling) while neighbors are energized to provide repulsive force (pushing) 370 mA current. Inter-actuator stroke ≈ 4 µm.

## Data Availability

All data needed to evaluate the conclusions in the paper are provided in the paper and/or the Appendix A. Additional data related to this paper may be requested from the authors.

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
