# Peer review of "A Design Approach to Reducing Stress and Distortion Caused by Adhesive Assembly in Micromachined Deformable Mirrors"

_micromachines, 2023, doi:10.3390/mi14040740_

Round 1

Reviewer 1 Report

The authors make use of St. Venant's principle to reduce high-spatial-frequency distortion from adhesive bonding by moving the bonding interface to the end of a long post.  The promise of the approach is demonstrated via FE modeling with parameters relevant to their application.  They then design an experiment and fabricate relevant structures from a SOI wafer using DRIE.  The results clearly show that the impact of adhesive distortions is greatly reduced by moving the adhesive interface away from the optical surface.

The paper is very well written.  The numerical and physical experiments are well designed and presented clearly.  The issue under investigation is relevant beyond actively deformable mirrors and of interest to a broad segment of the precision optics community.

The authors could have done parametric FE studies to show the dependence of distortion reduction as a function of several parameters and perhaps derive general design principles, but this could easily go beyond the scope of the presented work.  Perhaps this could be the subject of a future paper.

In Fig. 10 it could have been nice to show an actuation level that minimizes the rms surface height from Fig. 9, but this is not essential.

In general I congratulate the authors to their solid work and well written paper.  As a reviewer I appreciate it very much when authors submit papers that are publication-ready, which unfortunately is not the norm.  I recommend publication as is.

Reviewer 2 Report

In this work, the authors provide a novel synthetic approach for deformable mirrors, which aims to eliminate local deformations where actuators are adhered. Specifically, by bonding actuators to pillars on the silicon surface, instead binding directly to the silicon, the stress decays significantly along with local deformations due to adhesion. The authors provide simulation and experimental data showing that binding to pillars provides significant improvement, and fabricate a device using multiple actuators on pillars. Overall, this strategy seems sound to improve the performance of deformable mirrors, and the technique to do so is laid out well.

When the authors create a 37-magnet device, there is a significant deformation due to stress generated from the magnets. It's unclear initially what effect limiting local deformations has with this large amount of global deformation, though the authors failry state that high frequency deformations cannot be as easily corrected for by actuators. If such data is aailable, it may help to show a control device with 37-magnets bound directly to silicon to compare these results, showing local and global deformations. Otherwise, I recommend this work for publication.